# Cancer discrimination by on-cell N-glycan ligation

Shogo Nomura[1,2], Yasuko Egawa[1], Sayaka Urano[1], Tsuyoshi Tahara[3], Yasuyoshi Watanabe[3] &
Katsunori Tanaka [1,2,4,5✉]

In the field of molecular imaging, selectivity for target cells is a key determinant of the degree of imaging contrast. Previously, we developed a pre-targeted method by which target cells could be selectively imaged using a labeled N-glycan that was ligated in situ with an integrin-targeted cyclic RGD peptide on the cell surface. Here we demonstrate the power of our method in discriminating various cancerous and non-cancerous cells that cannot be distinguished using conventional RGD ligands. Using four cyclic RGDyK peptides with various linker lengths with five N-glycans, we identify optimal combinations to discriminate six types of $\alpha_v\beta_3$ integrin–expressing cells on 96-well plates. The optimal combinations of RGD and N-glycan ligands for the target cells are fingerprinted on the plates, and then used to selectively image tumors in xenografted mouse models. Using this method, various N-glycan molecules, even those with millimolar affinities for their cognate lectins, could be used for selective cancer cell differentiation.

[1] Biofunctional Synthetic Chemistry Laboratory, RIKEN Cluster for Pioneering Research, 2-1 Hirosawa, Wako, Saitama 351-0198, Japan. [2] GlycoTargeting Research Laboratory, RIKEN Baton Zone Program, 2-1 Hirosawa, Wako, Saitama 351-0198, Japan. [3] RIKEN Center for Biosystems Dynamics Research, Minatojima-minamimachi, Chuo-ku, Kobe, Hyogo 650-0047, Japan. [4] Biofunctional Chemistry Laboratory, A. Butlerov Institute of Chemistry, Kazan Federal University, 18 Kremlyovskaya street, Kazan 420008, Russian Federation. [5] Department of Chemical Science and Engineering, School of Materials and Chemical Technology, Tokyo Institute of Technology, 2-12-1 O-okayama, Meguro-ku, Tokyo 152-8552, Japan. ✉email: kotzenori@riken.jp

In vivo molecular imaging has attracted much attention as a method for analyzing the in vivo kinetics of anticancer drugs, their accumulation properties in target tissues, and expression levels in tumor, especially in the context of diagnostic applications (e.g., those using noninvasive fluorescent imaging, MRI, or PET). Although many promising tracers have been shown to target cancers, their selectivity and specificity could still be considerably improved. RGD peptides are high-affinity ligands of $\alpha_v\beta_3$ integrins, which are cell adhesion molecules that are highly expressed on tumor vasculature[1,2]. In addition to cancer-specific antibodies, RGD peptides have been applied to the molecular imaging of a wide range of cancers[3–5]. However, because integrins are also expressed on other endothelial tissues, RGD peptides often provide poor imaging contrast. RGD peptides nonselectively and strongly bind to various integrin-expressing tissues with $K_D$ values on the order of nM; moreover, they are rapidly captured by receptor-mediated endocytosis[6], which further decreases the signal ratio between tumor and background (Fig. 1a, the probe is depicted as a red circle). This problem arises frequently when using strongly interacting ligands, and it is very rare that a common receptor or antigen is expressed only on the target cell type. Alternatively, when applying a weakly interacting probe to a target receptor (e.g., a probe with a millimolar $K_D$)[7] even if the receptor is specifically expressed on target cells, the probe will not bind sufficiently strongly to enable efficient cell imaging (Fig. 1b, blue triangle).

Recently, we proposed an approach for distinguishing normal HUVECs (human umbilical vein endothelial cells) from cancerous HeLa cells (human cervical cancer cells), both of which express common receptors such as $\alpha_v\beta_3$ integrins, by combining an RGD peptide with a glycan in a pre-targeted manner (Fig. 1c)[8]. In this approach, a cyclic RGDyK peptide with high affinity for $\alpha_v\beta_3$ integrins, prepared with an azide tag instead of an imaging label, was initially pre-targeted to both HUVECs and HeLa cells. Subsequently, the cells were treated with α(2,6)sialyl-N-glycan with both a fluorescent label and dibenzocyclooctyne (DBCO), which reacts bioorthogonally with azide. This glycan has only

weak affinity for PECAM (sialic acid–binding protein), which is selectively expressed on HUVECs; weak binding to the pre-targeted HUVECs selectively anchored the fluorescence to the cell through the strain-promoted azide-alkyne cycloaddition (SPAAC) reaction, which was facilitated by the proximal effects between the two surface receptors. The synergy between the strong and weak ligand/receptor interactions and in situ ligation on the cell surface enabled highly selective imaging of the HUVECs. It should be noted that cell imaging using pre-linked RGD-glycan conjugates did not produce significant imaging contrast because the interactions between the pre-linked molecules and cell surface were dominated by strong RGD/integrin interactions. Therefore, the in situ ligation concept is very important for exploiting the advantages of both the strong and weak interactions in order to achieve clear cell imaging contrast. As we have investigated in the previous research[8], the clicked products are efficiently internalized with the same efficiency as RGD peptide, while glycans are less efficiently internalized. Because the click reaction should occur on the cell surface, click ligation and subsequent internalization could be the key of the method to image the target cells.

In demonstrating the power of our pre-targeting strategy to selectively target and image the cancer cells, in this paper we discriminate cancerous and non-cancerous cells that all express $\alpha_v\beta_3$ integrins on their cell surface. While the conventional cyclic RGDyK peptide, as a "strongly interacting ligand", cannot selectively image the targeted cells, our method discriminates the cancer cells both in vitro and in vivo depending on the N-glycan structures as "weakly interacting ligands" with specific lectins and on the spatial arrangement between the ligand/receptor complexes on the target cell surface.

For this study, we select six cell lines: HeLaS3 (human cervical cancer cells)[9], A549 (adenocarcinomic human alveolar basal epithelial cells)[10], BxPC3 (human pancreatic cancer cells)[11], PC3 (human prostate cancer cells)[12], and SW620 (human colon cancer cells)[13] as cancerous cells, and TIG3 (human fibroblast cells)[14] as a non-cancerous cell. All of these cell lines express $\alpha_v\beta_3$ integrins on their surface; therefore, if successful, our pre-targeting approach could be generalized.

## Results

**Design of cyclic RGD peptides and N-glycan ligands**. Both functionalized RGD peptide (strongly interacting ligands) and N-glycans (weakly interacting ligands) were designed and prepared according to a previously reported method (Fig. 2 and Supplementary Methods)[8]. Thus, the azide function was introduced onto the cyclic RGDyK peptide ligands (Fig. 2, **1a–d**) through various lengths of polyethylene glycol (PEG) linkers, i.e., PEG3, PEG5, PEG7, or PEG9, in order to investigate the distance dependence of the click reaction, which dictated the spatial arrangement between the ligand/receptor complexes that was tolerated in the context of the labeling reaction. To differentiate cancer cells on the basis of specifically expressed lectins, five biantennary N-glycans terminated by either N-acetylneuraminic acids **2a,b** [either α(2,6)- or α(2,3)-linked to galactose], galactose **2c**, N-acetylglucosamine **2d**, or mannose **2e** were applied. Both fluorescent labels, tetramethylrhodamine (TAMRA) and DBCO, were attached to the reducing ends of the glycan molecules through a short PEG linker.

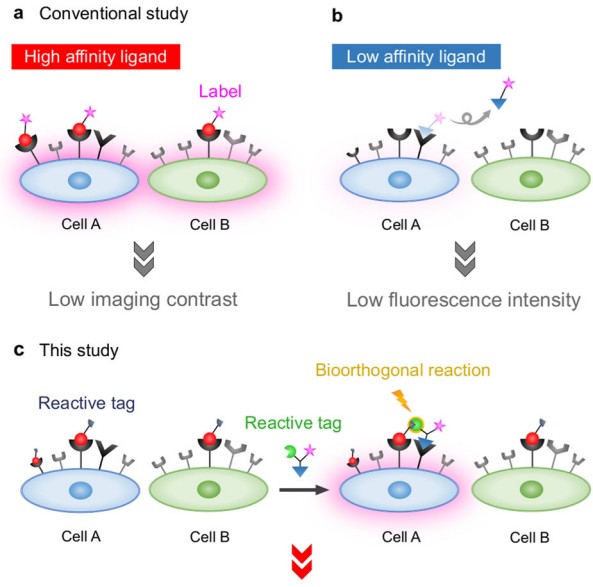

**Fig. 1 Schematics of cell imaging methods involving high- and low-affinity ligands for cell surface receptors. a, b** Conventional labeling using only a high-affinity ligand (**a**) or a low-affinity ligand (**b**). **c** Labeling using both high- and low-affinity ligands and a bioorthogonal reaction on the cell surface.

**Discrimination of various cancerous and non-cancerous cells**. With the functionalized RGD peptide ligands and N-glycan ligands in hand, we demonstrated selective targeting of two cell-surface receptors by directly linking their high- and low-affinity ligands in situ (Fig. 3). In order to rapidly identify the optimal

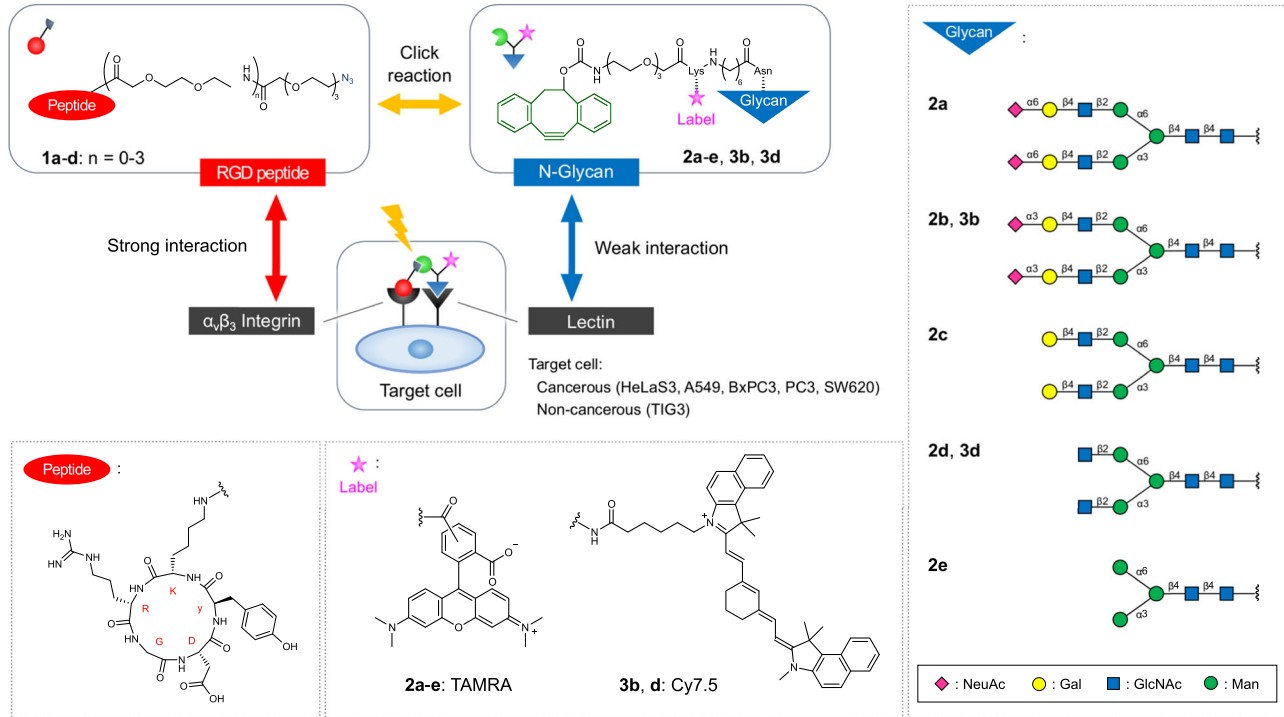

**Fig. 2 Structures of the functionalized cyclic RGDyK peptides and N-glycan ligands.** An azide was attached to the cyclic RGDyK peptide thorough various length of PEG linkers. Both fluorescent labels and dibenzocyclooctyne, which react selectively with the azide, were attached to the reducing ends of the glycan molecules. Glycans are shown as symbols.

combination of both RGD and N-glycan ligands for imaging of the target cancer cells, the four RGD peptides and the five glycan ligands were screened simultaneously in a library fashion on cells in 96-well plates. We applied our previously optimized cell incubation conditions for pre-targeting and the strain-promoted click reaction to ensure that the azide group on the integrin-targeted cyclic RGDyK had a sufficiently high clickable reactivity on the cell surface[8]. Namely, the cells ($5 \times 10^4$) in each well were initially treated with the cyclic RGDyK peptides **1a–d** (50 μM) for 15 min at room temperature (RT). After the cells were washed with medium to remove excess peptide, the pre-targeted cells were further incubated with N-glycan ligands **2a–e** (100 μM) for 30 min at 4 °C. At room temperature, the higher rate of integrin-mediated internalization of the on-cell cyclic RGDyK reduced the efficiency of the on-surface click reaction when compared to the 4 °C reaction conditions (Supplementary Figs. 16 and 17)[8]. After the cells were washed with medium, they were fixed with 4% paraformaldehyde prior to fluorescent microscopy analysis. As control experiments, cells were treated with (1) cyclic RGDyK ligand labeled with TAMRA or (2) only glycan ligands **2a–e**.

As expected, the cyclic RGDyK ligand labeled with TAMRA fluorescence, which has been widely used to visualize $\alpha_v\beta_3$ integrins on tumor cells, could be used to image all cell lines very clearly, but could not discriminate among them (Fig. 3, upper field). Alternatively, the N-glycan ligands **2a–e** without the pre-targeting procedure did not label any of the cell lines (Fig. 3, left vertical lines).

On the other hand, a combined use of cyclic RGDyK peptides and N-glycan ligands resulted in specific cell labeling, dependent on N-glycan structures, namely, GlcNAc- and Man-terminated glycans (**2d,e**) on HeLaS3 cells (Fig. 3a), α(2,3)Sia-terminated glycan (**2b**) on A549 cells (Fig. 3b), Gal-, GlcNAc-, and Man-terminated glycans (**2c,d,e**) on BxPC3 cells (Fig. 3c), α(2,6)Sia-, α(2,3)Sia-, and GlcNAc-terminated glycans (**2a,b,d**) on SW620 cells (Fig. 3e), and α(2,6)Sia-, α(2,3)Sia-, GlcNAc-, and

Man-terminated glycans (**2a,b,d,e**) on TIG3 cells (Fig. 3f). No combinations of the N-glycans used in this research targeted PC3 cancer cells (Fig. 3d); nevertheless, such fingerprints of "pre-targeting" cell labeling profiles could discriminate among these six cell lines (five cancerours, one non-cancerous cell line), which are very difficult to distinguish using commonly applied 'strongly interacting' RGD peptides.

We also noted that not only the structures of N-glycans, but also the linker lengths of the RGD peptide connected to the azide function, could affect the cell labeling profiles. For instance, BxPC3 cells were only significantly labeled by the Man-terminated glycan (**2e**) when pre-targeted with the RGD derivative **1c** (Fig. 3c). Similarly, non-cancerous TIG3 cells were only stained by the GlcNAc- or Man-terminated glycans (**2d** and **e**) when combined with RGD **1c** (Fig. 3f). These results may dictate the optimal spatial arrangement between the cyclic RGDyK peptides and specific N-glycans in order to be "clicked" onto the target cell surface, and hence the ideal arrangement between the $\alpha_v\beta_3$ integrins and target lectins. As such, the cell labeling fingerprints obtained in Fig. 3 represent not only the information derived from two cell-surface interactions of "high" and "low" affinity ligands, but also that of the spatial arrangement between two receptors (or receptor/ligand complexes). Thus, even cells that are very similar in terms of expressing the same two receptors on their surface could be differentiated on the basis of the cell surface arrangement. Furthermore, such receptor profiles could be used to identify each cell line, if necessary.

Previously reported lectins expressed on these six cell lines are summarized in Table 1. Although few lectins have been investigated on non-cancerous TIG3 cells, the specific cell surface lectins in Table 1 represent potential candidates for the respective N-glycans identified in Fig. 3, e.g., vimentin and SP-D for GlcNAc- and Man-terminated glycans (**2d,e**) for targeting HeLaS3 cells.

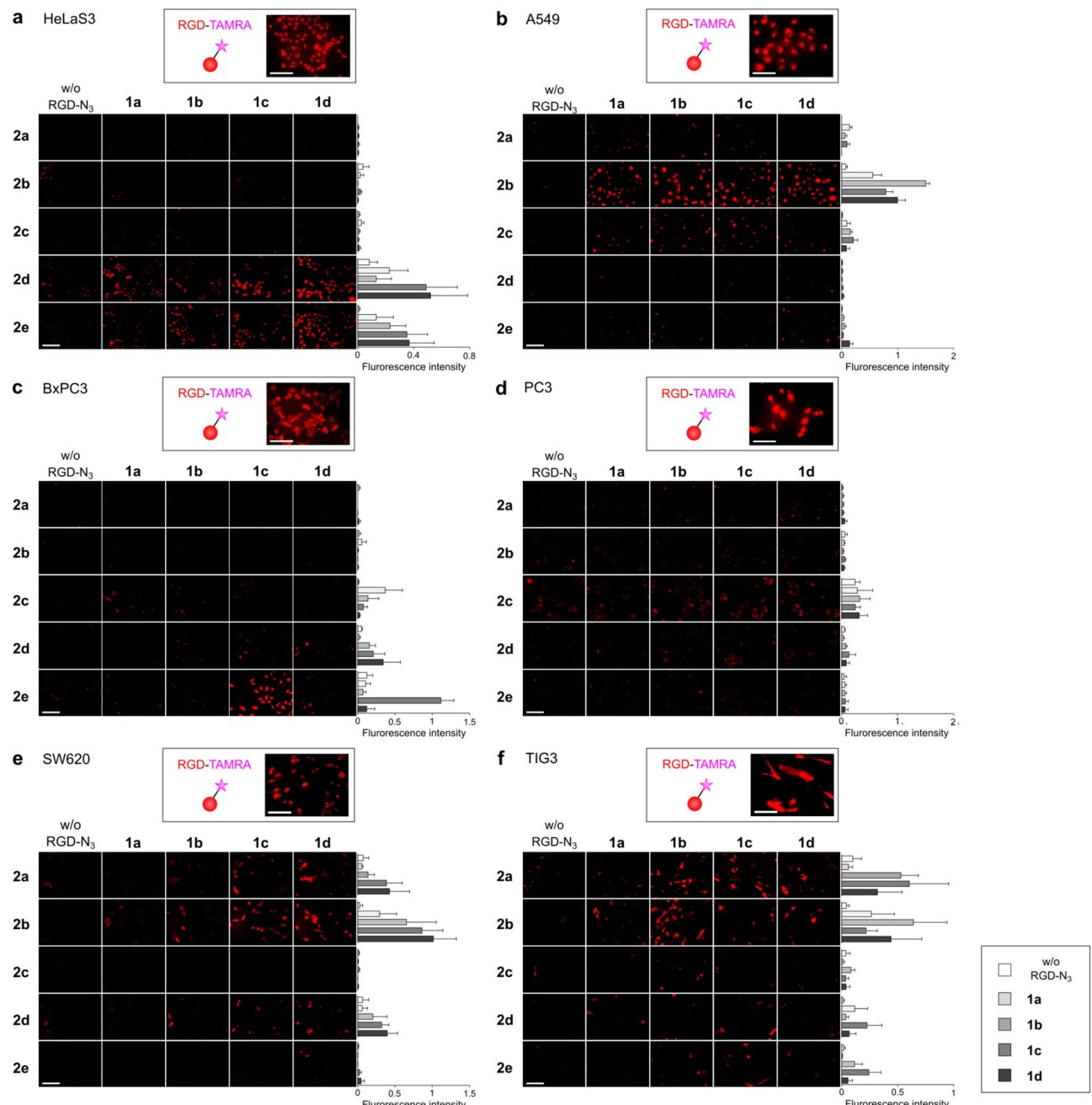

**Fig. 3 Fingerprints providing the optimal cyclic RGDyK and N-glycan ligands for specific cell labeling.** Specific cell types [HeLaS3 (**a**), A549 (**b**), BxPC3 (**c**), PC3 (**d**), SW620 (**e**), and TIG3 (**f**)] were seeded on 96-well plates ($5 \times 10^4$ cells/well) and treated with one of four cyclic RGDyK ligands (50 μM, RT, 15 min) or one of five fluorescently labeled N-glycan ligands (100 μM, 4 °C, 30 min). Fluorescence intensities were normalized against cell number ($n = 4$ plates). Imaging with simple TAMRA-cyclic RGDyK ligand is provided in the upper field of each cell fingerprint, **a–f** Scale bar, 100 μm. Data presented as mean ± SEM.

**Table 1 Literature survey of lectins expressed on cancerous and non-cancerous cells, as well as their glycan specificities.**

| Cell Lines | Lectin expression and their known ligands | | | | |
|---|---|---|---|---|---|
| | **2,6-Sia** | **2,3-Sia** | **Gal** | **GlcNAc** | **Man** |
| HeLa | Siglec-3[16] | Galectin-1[17] | – | Vimentin[18] | SP-D[19] |
| A549 | Siglec-10[20] | Galectin-1[21] | – | – | SP-D[22] |
| BxPC3 | – | – | Galectin-3[23] | Vimentin[24] | DC-SIGN[25] |
| PC3 | – | Galectin-1[26] | – | – | – |
| SW620 | Siglec-7[27] | Galectin-8[26] | – | Vimentin[28] | – |
| TIG3 | – | – | – | Vimentin[29] | – |

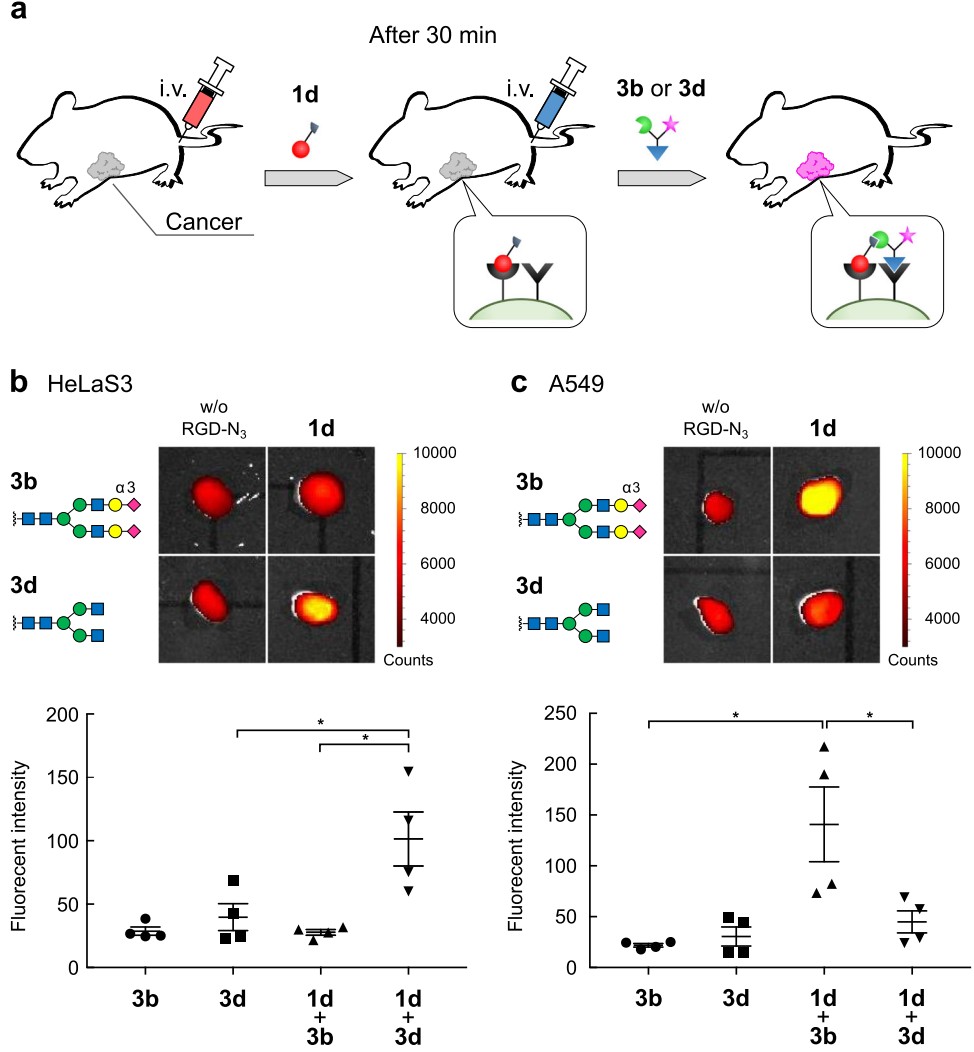

**Fig. 4 Selective labeling of cancer tissues in a mouse xenograft model. a** Mice ($n = 4$) were treated intravenously with RGD peptide ligand **1d** (150 nmol in 100 μL saline), and then after 30-min intervals, with N-glycan ligands **3b** and **3d** labeled with the near-infrared fluorescent dye Cy7.5 (15 nmol in 100 μL saline). After dissection, tumor regions [HeLaS3 (**b**) and A549 (**c**)] were imaged on a PerkinElmer IVIS Spectrum in vivo imaging system (ex: 640 nm; em: 710 nm). Data presented as mean ± SEM, *$p < 0.05$.

**Selective cancer targeting in vivo.** Based on the cell labeling profiles (i.e., cell-identifying fingerprints) obtained in Fig. 3, we investigated whether our pre-targeting protocol could also be applied to discriminate cancers in mice (Fig. 4). HeLaS3 and A549 cancer cells were selectively targeted by GlcNAc-terminated glycan (**2d**, Fig. 3a) and α(2,3)Sia-terminated glycan (**2b**, Fig. 3b) respectively, upon treatment with cells pre-targeted by cyclic RGDyK peptide **1d**. To clearly analyze the fluorescence signals derived from the pre-targeting protocol among the various biologically relevant fluorescence backgrounds from mouse tissues, we used the near-infrared fluorescence dye, Cy7.5 (see structure in Fig. 2). The amount of the RGD and N-glycan ligands administered to each mouse were adjusted to the amounts used in the cell experiments shown in Fig. 3, assuming ~2 mL of blood volume, so that the pre-targeting and receptor-mediated bioorthogonal click reaction could proceed in tumor regions even in mouse models. Thus, BALB/c nude mice implanted with either HeLaS3 or A549 cells in the left shoulder regions were initially treated with cyclic RGDyK peptide **1d** (intravenous injection, 150 nmol). After 30-min intervals, N-glycan ligands **3d** and **b** (15 nmol) were intravenously administrated. After 3 h, the mice were dissected, and the fluorescence of cancer regions was analyzed on an IVIS Spectrum in vivo imaging system (Fig. 4a).

The fluorescently labeled glycans themselves accumulated more or less in the tumor regions non-specifically (largely due to the primarily and hydrophobic fluorophore structures), that in turn caused the increase of background fluorescence. However, gratifyingly, both HeLaS3 and A549 tumors were fluorescently surface-labeled to greater extent, i.e., by more than 2.5-fold increase, by injection of injecting both functionalized cyclic RGDyK and the "matched" N-glycan ligands than by injection with RGDyK or N-glycan alone (i.e., without the pre-targeting procedure) or by non-targeted glycan combinations (Fig. 4b, c). These data clearly validate our pre-targeted strategy for discriminating cancers even in vivo.

## Discussion

In our previous proof-of-concept research, we imaged the PECAM-expressing cells by efficiently linking the integrin to PECAM on a cell surface by two cell surface ligands, i.e., RGD and labeled sialo-N-glycan. The cell surface orientation and conformation of two receptors, i.e., PECAM and integrin, are

dynamic, hence their ligands could react each other when receptors are close enough on the ligand binding. While crystallographic data of PECAM or detailed information of integrin/ PECAM complex on cell surface are not currently available, the proximal information has been discussed based on the experimental evidences that the microscopic analysis of two receptors detected their co-localization on a cell surface[15]. Thus, an attractive feature of our approach is that this method may be used to directly analyze the relative spatial arrangements of two and possibly more receptors on a cell surface, in dynamic fashion, by using linkers of various lengths. Cell surface dynamics, as indicated by the spatial arrangements of the target proteins and, hence, the ligand-directed signaling pathways, could be monitored or even controlled simply by treating the live cells with chemical reagents and subsequently imaging.

Based on the Table 1, A549 express the SP-D, but the Man-terminated glycan **2e** could not image the cells. Alternatively, the galactose-terminated glycan **2c** slightly stained the PC3 cells regardless of the linker length of RGD derivatives **1a–d**, while no representative mannose-binding lectins are expressed on the PC3 cell surface. This could be rationalized by the spatial arrangement and distance between the integrin and lectins on cell surface. In addition, the cell surface binding of glycans, especially when the interaction is very weak, is not only caused by the glycan/lectin interaction, but also by the hydrophobic or H-bonding interactions with various biomolecules on cell surface. The multiple interactions by the glycan on the cell surface complicate the analysis and the result cannot simply be rationalized by the lectins expressed on the cell surface.

We also checked the "Human Protein Atlas" and "RefEx" databases (Supplementary Table 2), but the mRNA expression data are not in consistent with those provided in Table 1. For examples, among the currently available mRNA data for three cell lines (HeLa, A549, and PC3), SP-D in HeLa, which is expected to bind to Man-terminated glycan **2e** by Table 1 and then experimentally validated in Fig. 3a, cannot be rationalized from these mRNA-based databases (Supplementary Table 2). In Table 1, we summarized the previous literatures describing the cell surface lectins and their targeted glycans, which were analyzed by biochemical methods. Although the glycan interaction is not necessarily explained by the surface lectins, e.g., by interaction with membrane proteins, glycans, or lipids, the comparison of our cell surface chemistry with previously reported cell surface lectin analysis in literature may provide quite promising directions for future cell targeting or imaging.

In summary, we could discriminate each type of $\alpha_v\beta_3$ integrin–expressing cancerous and non-cancerous cells by applying both the strongly interacting cyclic RGD and weakly interacting N-glycan ligands of the surface receptors and ligating them in situ on the cell surface. Cell-identifying profiles were efficiently screened on 96-well plates, and the optimal RGD peptide and glycan ligands were imaged as "fingerprints" on the plates. It should be noted that our cell-identifying fingerprints profile two cell surface receptors/ligands interactions, as well as their spatial orientation. These fingerprints were then used to selectively image the targeted cancers in xenografted mice. Our proof-of-concept studies described in this paper could potentially be applied not only to the glycan/lectin interactions, but also to any other "weak" interactions on the cell surface. Previously, these types of interactions have been neglected and hence underutilized. Various low-affinity ligands can now be used for cell discrimination and selective imaging research.

## Methods

**General**. N-Glycan derivatives were supplied from GlyTech Inc. (Kyoto, Japan). Boc-miniPEG™, Boc-miniPEG-3™, and EDC·HCl were obtained from Peptide Institute, Inc. (Osaka, Japan). 5-(and-6)-Carboxytetramethylrhodamine, succinimidyl ester (5(6)-TAMRA, SE) was obtained from AAT Bioquest (California, USA). 3-(5-Carboxypentyl)-1,1-dimethyl-2-((E)-2-((E)-3((E)-2-(1,1,3-trimethyl-1H-benzo[e]indol-2(3 H)-ylidene)ethylidene)cyclohex-1-enyl)vinyl)-1H-benzo[e] indolium (Cy7.5) chloride was obtained from abcam (Cambridge, UK). All other chemicals and solvents of special grade were obtained from Tokyo Chemical Industry, co., Ltd (Tokyo, Japan) or Wako Pure Chemical Industries, Ltd (Osaka, Japan), and were used without purification. HPLC was performed on Shimadzu liquid chromatograph CBM-20A, LC-20AD, and SPD-20AV (Kyoto, Japan) with an analytical column COSMOSIL $_5C_{18}$-AR-300 (4.6 mm ×250 mm, Nacalai Tesque, Inc., Kyoto, Japan) at a flow rate of 1 mL/min, and on JASCO liquid chromatograph LC-NetII/ADC, PU-2089 Plus, and UV-2075 Plus (Tokyo, Japan) with a preparative column COSMOSIL $_5C_{18}$-AR-300 (20 mm × 250 mm, Nacalai Tesque, Inc.) at a flow rate of 7 mL/min. Mass spectra were recorded on a Bruker micrOTOF QIII (Rheinstetten, Germany). NMR spectra were recorded on a JEOL AL400 spectrometer (Tokyo, Japan).

**Synthetic procedures**. See Supplementary Methods. For HPLC charts see Supplementary Figs. 1–15. For NMR spectra see Supplementary Fig. 20.

**Cell culture**. The six cell lines used in this study were obtained from either American Type Culture Collection (Virginia, USA), RIKEN Cell Bank, or JCRB Cell Bank. HeLaS3, A549, BxPC3, PC3, and TIG3 cells were cultured in DMEM (Wako) containing 10% fetal bovine serum (FBS, Wako) and penicillin-streptomycin (Thermo Fisher Scientific, Inc., Massachusetts, USA). SW620 cells were cultured in Leibovitz's L-15 Medium (Wako) containing 10% FBS and penicillin-streptomycin. We check the mycoplasma contamination by MycoAlert™ mycoplasma detection kit (Lonza Walkersville, Inc., USA) regularly and also before performing this research.

**In vitro imaging studies**. HeLaS3, A549, BxPC3, PC3, TIG3, and SW620 cells were seeded onto 96-well plates. A solution of the cyclic RGDyK peptide ligands **1a–d** (50 μM, 100 μL) in medium was added to the cells, and the solution was incubated for 15 min at room temperature. As a positive control, a solution containing only TAMRA-labeled cyclic RGDyK peptide[8] was also incubated at same concentration. After washing the cells twice with medium, the cells were treated with a solution of glycan ligands **2a–2e** (100 μM, 25 μL) in medium for 30 min at 4 °C. The cells were then washed with medium, fixed with 4% paraformaldehyde in PBS for 10 min, and labeled with Hoechst33342 (Thermo Fisher Scientific, Inc.) to count cell number. The samples were imaged using an BZ-X 700 (Keyence, Osaka, Japan). Fluorescent intensity was calculated by ImageJ (U. S. National Institutes of Health, Maryland, USA), and the scores were normalized by cell number. The labeling experiments on the plate were performed for four times each cell lines, and data was averaged (Supplementary Table 1).

**Laboratory animals**. In all, 8–10-week-old BALB/c-nu/nu mice were purchased from CLEA Japan (Tokyo, Japan). The mice were housed at the RIKEN Center for Biosystems Dynamics Research (BDR). All injections to the mice were performed under anesthesia. All procedures involving mouse experiments were approved by the Ethics Committee of RIKEN (MAH21-19-17), and performed in accordance with the institutional and national guidelines.

**Selective cancer pre-targeting in vivo**. The cancer model mice were prepared by subcutaneously injecting HeLaS3 ($2 \times 10^6$ cells/100 μL) or A549 ($4 \times 10^6$ cells/100 μL) into the left shoulder of 8–10-week-old BALB/cAJc1-nu/nu mice. The mice injected HeLaS3 or A549, were housed for 3 weeks or 4 weeks, respectively. The solution of cyclic RGDyK peptide **1d** (150 nmol/100 μL saline) was injected into mice via tail vein. After 30 min, the solutions of glycan ligands **3b** or **3d** (15 nmol/ 100 μL saline) were injected. After 3 h, the mice were dissected, and the fluorescence of cancer region was analyzed on an IVIS Spectrum in vivo imaging system (Caliper Life Sciences Inc, Massachusetts, USA, $n = 4$) see Supplementary Figs. 18 and 19.

## Data availability
All data supporting the findings of this study are available with the article, and can also be obtained from the corresponding author upon reasonable request. The source data underlying Figs. 3 and 4 are provided as Supplementary Data 1.

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

## Acknowledgements

This work was supported by JSPS KAKENHI Grants-in-Aid for Scientific Research on Innovative Areas ("Middle Molecular Strategy") Numbers JP16H03287, JP18K19154, and JP15H05843. This work was also performed with the support of the Russian Government Program for Competitive Growth, awarded to Kazan Federal University. We would also like to thank GlyTech Inc. for supplying various N-glycans.

## Author contributions

Preparation of reagents was done by S. N. Cell imaging studies were carried out by S.N. and Y.E. In vivo imaging studies were done by S.U. and T.T. The manuscript was written by S.N. and K.T. and checked by Y.W. This research was directed and supervised by K.T.

## Competing interests

The authors declare no competing interests.
