## [Peer Review File · Communications Chemistry]

Reviewers' comments:

Reviewer #1 (Remarks to the Author):

The manuscript "Cancer discrimination by "on-cell" N-glycan ligation" by Nomura et al. depicts the development of a dual-strategy to specifically label cells based on the presence of two cell surface receptors. The approach targets both high-affinity (integrin-RGD) and low-affinity (lectin-glycan) interactions to boost both labelling intensity and specificity. Cells are pre-incubated with cyclic RGD harboring an azide tag before being treated with fluorescently labeled, DBCO-charged derivatives of five different N-glycans. Specificity is achieved in the combination of different glycans and linker lengths employed, and the approach is applied in an ex vivo tumor imaging model.

Despite not being the first example for this approach, the work is interesting and novel. Particularly, the strategy could be extended to generate cell-specific "lectome" data once more probes are available. Several issues remain with the work and the manuscript that should be sorted out before any recommendation can be made:

- The microscopy images in Fig. 2 seem to have a considerably higher background whenever positively labeled cells are seen. Microscopy settings should be equal for all images in one experiment (e.g. all images taken with the same cell line), and should be treated the same way to avoid misinterpretation. Furthermore, the histograms in Fig. 2 should be made more obvious, as some are hard to see.
- Throughout the manuscript, treating the cell lines used as "cancer" vs "normal" is somewhat confusing. First, "normal" should be replaced with "non-cancerous". Second, the cell type of TIG-3 cells should be depicted as fibroblasts. And in the absence of a matched cancer cell line as a control, TIG-3 should be considered (and discussed) as another example for a cell line instead of the "normal" control line.
- The ex vivo work is nice, but could the authors please state what the reason for the high background fluorescence is even without RGD peptide? In Figure S16, non-treated control tumors do not exhibit any background, while treatment with glycans alone generates substantial fluorescence. It may be good to see intravital images of the whole mouse if available. Furthermore, an intensity reference should be included in Fig. 4a.
- Glycans should be characterised by NMR either in their commercial form or after reaction to 2a-e. Due to the low amounts available, a ¹H NMR should be enough.
- Cell lines should be tested for mycoplasma contamination - as contamination could potentially lead to modulation of lectin expression, contamination has to be ruled out to ensure reproducibility of the results.
- Lectin expression in Table 1 could be cross-checked and validated with the expression levels depicted in the Human Protein Atlas if applicable.
- In figure legends, the nature of the plot and of the error bars (mean/median, SEM/SD) should be mentioned. Histograms in Fig. 4 should be swapped for dot plots containing individual data points, as n=4.
- The authors mention missing internalisation as an advantage of their method. Could they please comment on the capacity for internalisation of the covalent adduct between RGD and glycan? Is an endosomal signal usually seen?
- The dependence of labeling on the linker length between peptide and glycan is surprising and interesting. Could the authors please further comment on this, maybe with an example on the

molecular weight/size of a particular lectin and the distance to integrins?

Furthermore, a few typographical errors were found:

- Throughout the manuscript, italic "N" is used in GlcNAc, N-glycan etc. These should be non-italic.
- In lines 85 & 86 "NAc-glucosamine" and "NAc-neuraminic acids" are not the official nomenclature for these monosaccharides.
- Line 26, "a great deal of attention" should be re-phrased.
- Line 88, "through an appropriate linker" should be re-phrased to a more specific sentence.
- Line 105, "...the cyclic RGDyK ligand (...) could image all cells..." should be re-phrased to "...could be used to image all cells...".
- Line 148, "biorthogonal" was probably autocorrected from "bioorthogonal"
- Line 155, "to a greater extent" should be re-phrased and contain the fold increase of labeling.
- Line 155, "by injection of injecting" word duplicated.
- Line 193, "Scare bar" should be "Scale bar".
- Line 241, "Hoechest" should be "Hoechst".

Reviewer #2 (Remarks to the Author):

In the manuscript titled Cancer discrimination by "on-cell" N-glycan ligation by Nomura et al., the authors describe their further development of a technique where cancer cells are specifically labeled using a click reaction between a high-affinity ligand and a low-affinity ligand. These ligands are brought together on the cell surface with the assistance of a combination of $\alpha\beta3$ integrin and lectin unique to a particular cell type. The technique uses the strong affinity between $\alpha\beta3$ integrin and a cyclic RGDyK peptide ligand to pre-target cancer cells. The cyclic RGDyK peptide ligand is functionalized with an azide that is tethered using various ethylene glycol linker lengths. Subsequently, a low-affinity N-glycan ligand bearing a fluorescent group and a dicyclobenzooctyne (DIBO) is added. A weak interaction between the N-glycan ligand and a cell-surface-expressed lectin recruits the second reagent to the cell surface, where it is in close enough proximity to react with the RGD-azide ligand. Five different N-glycan variations were used.

The authors have shown this technique using HeLa and HUVEC cells in a previous publication. They noted in their previous publication that the ethylene glycol linker length has an affect on the labeling efficiency. In the current manuscript, the authors use five cancer cell lines and one normal cancer cell line, and measure fluorescence intensity using different combinations of cyclic RGDyK peptide linker length and N-glycan ligand. The combination-dependent labeling results in a fingerprint pattern specific to a cell type. In addition, the authors show that the reagents can be used to label tumors from mouse xenograft models using matched pairs of functionalized cyclic RGDyK ligand and N-glycan ligand to show selective labeling of a specific tumor type.

Finally, they suggest that other weak interactions can be used.

Verdict:

I recommend this manuscript for publication following minor revisions.

Comments and questions:

The authors present a novel tool for enhancing weak interactions specific to a cell type. I imagine this system like a broadly applied primer or undercoat that improves the adhesion and durability of the precisely applied paint. I would like to see the expression levels of the cell surface proteins and compare them to the labeling efficiency. Although unexpectedly low labeling could be due to

spatial arrangement, protein expression levels could be playing a role in labeling efficiency. One may consider that spatial arrangement may also have less effect if the labeling is performed at room temperature instead of 4°C, i.e., the cell-labeling profiles may be temperature dependent. I agree with the authors' sentiment that this concept can be used for exploring other weak interactions and has potential for use in therapeutic delivery as well.

For the sentence:

On the other hand, a combined use of cyclic RGDyK peptides and N-glycan ligands resulted in specific cell labeling...Gal- and GlcNAc-terminated glycans (2c,d) on BxPC3 cells (Fig. 3c),... 2e should be added as well.

Also:

... $\alpha(2,6)$ Sia- and $\alpha(2,3)$ Sia-terminated glycans (2a,b) on normal TIG3 cells (Fig. 3f). 2d,e should be added as well.

Is spatial arrangement the reason that we don't see binding of 2e to A549 or is it protein levels? Should additional linker lengths be explored?

Why does 2c appear to bind to PC3?

TIG3 shows strong staining with 2a,b but only has Vimentin. BxPC3 also expresses Vimentin, but is not stained with 2a,b. Vimentin is a mesenchymal marker and its expression level is dependent on the epithelial or mesenchymal characteristic of the cell. One may deliberately induce epithelial-to-mesenchymal transition, verify upregulated Vimentin, and test this pre-targeting technique again.

Referee: 1

1) The microscopy images in Fig. 2 seem to have a considerably higher background whenever positively labeled cells are seen. Microscopy settings should be equal for all images in one experiment (e.g. all images taken with the same cell line), and should be treated the same way to avoid misinterpretation. Furthermore, the histograms in Fig. 2 should be made more obvious, as some are hard to see.

We apologize that our figure presentation led to the misunderstanding of the reviewer. We previously highlighted the critical images (discussed in this manuscript) with the red marker. We deleted the red highlights and enlarged the images. Of course, we performed all cell experiments by setting the same conditions for microscopic analysis.

2) Throughout the manuscript, treating the cell lines used as "cancer" vs "normal" is somewhat confusing. First, "normal" should be replaced with "non-cancerous". Second, the cell type of TIG-3 cells should be depicted as fibroblasts. And in the absence of a matched cancer cell line as a control, TIG-3 should be considered (and discussed) as another example for a cell line instead of the "normal" control line.

Thanks to the reviewer's comments, we replaced "normal" with "non-cancerous". We also described TIG-3 as "fibroblasts" and discussed as another example of a cell line.

3) The ex vivo work is nice, but could the authors please state what the reason for the high background fluorescence is even without RGD peptide? In Figure S16, non-treated control tumors do not exhibit any background, while treatment with glycans alone generates substantial fluorescence. It may be good to see intravital images of the whole mouse if available. Furthermore, an intensity reference should be included in Fig. 4a.

For mice experiments, the fluorescently labeled glycans themselves accumulated more or less in the tumor regions non-specifically (largely due to the big and hydrophobic fluorophore structures), that in turn caused the increase of background fluorescence. These problems could not be avoided in any in vivo fluorescent imaging studies, but the main focus of research is detecting the differences in fluorescence intensity among the tumors depending on the glycan structures. We mentioned this in page 10, line 161-163. We added the representative whole body images of mouse, treated with RGD **1d** and glycan **3b** or **3d**, in Figure S17. We also added the fluorescence intensity reference in Fig. 4a.

4) Glycans should be characterised by NMR either in their commercial form or after reaction to 2a-e. Due to the low amounts available, a 1H NMR should be enough.

To describe the quality of N-glycan molecules used in this study, we commented the preparation and purity in Supporting Material Section 1, and also provided the NMR spectrum of the starting glycans in Supporting Material Section 5.

5) Cell lines should be tested for mycoplasma contamination - as contamination could potentially lead to modulation of lectin expression, contamination has to be ruled out to ensure reproducibility of the results.

We check the mycoplasma contamination regularly and also before performing this research. We commented this in page 21, line 273-274.

6) Lectin expression in Table 1 could be cross-checked and validated with the expression levels depicted in the Human Protein Atlas if applicable.

We appreciate the reviewer's constructive comments and suggestions. We checked the "Human Protein Atlas" and "RefEx" databases, but the mRNA expression data are not in consistent with those provided in Table 1. For examples, among the currently available mRNA data for three cell lines (HeLa, A549, and PC3, see new Table S2), SP-D in HeLa, which is expected to bind to mannosyl glycan **2e** by Table 1 and then experimentally validated in Figure 3a, cannot be rationalized from these mRNA-based databases in Table S2. In this research, we throughout investigated the previous literatures describing the cell surface lectins and their targeted glycans, which were analyzed by biochemical methods, and summarized in Table 1. Although the glycan interaction is not necessarily explained by the surface lectins, e.g., by interaction with membrane proteins, glycans or lipids (see below), the comparison of our cell surface chemistry with previously reported cell surface lectin analysis in literature may provide quite promising directions for future cell targeting or imaging. We added the new supporting Table S2 in Supporting Material and commented in page 11-12, line 195-205.

7) In figure legends, the nature of the plot and of the error bars (mean/median, SEM/SD) should be mentioned. Histograms in Fig. 4 should be swapped for dot plots containing individual data points, as n=4.

Thanks to the comments, we corrected the presentation of all figures.

8) The authors mention missing internalisation as an advantage of their method. Could they please comment on the capacity for internalisation of the covalent adduct between RGD and glycan? Is an endosomal signal usually seen?

As we have investigated in the previous research (*Adv. Sci.*, **2017**, 1700147), the clicked products efficiently internalized with the same efficiency as RGD peptide, while glycans hardly internalized. Because the click reaction should occur on the cell surface, click ligation and subsequent internalization could be the key of the method to image the target cells, as this reviewer commented. We commented this in page 4-5, line 62-66.

9) *The dependence of labeling on the linker length between peptide and glycan is surprising and interesting. Could the authors please further comment on this, maybe with an example on the molecular weight/size of a particular lectin and the distance to integrins?*

Thank you for the constructive comments and suggestion. In our previous proof-of-concept research, we imaged the PECAM (platelet endothelial cell adhesion molecule)-expressing cells by efficiently linking the integrin to PECAM on a cell surface by two cell surface ligands, i.e. RGD and labeled sialo-N-glycan. The cell surface orientation and conformation of two receptors, i.e., PECAM and integrin, are dynamic, hence their ligands could react each other when receptors are close enough on the ligand binding. While crystallographic data of PECAM or detailed information of integrin/PECAM complex on cell surface are not currently available, the proximal information has been discussed based on the experimental evidences that the microscopic analysis of two receptors detected their co-localization on a cell surface (Wong, C. W. Y. et al. PECAM-1/CD31 Trans-homophilic Binding at the Intercellular Junctions Is Independent of Its Cytoplasmic Domain; Evidence for Heterophilic Interaction with Integrin α v β 3 in *Cis. Mol. Biol. Cell* 11, 3109-3121 (2000)). Thus, an attractive feature of our approach is that this method may be used to directly analyze the relative spatial arrangements of two and possibly more receptors on a cell surface, in dynamic fashion, by using linkers of various lengths. Cell surface dynamics, as indicated by the spatial arrangements of the target proteins and, hence, the ligand-directed signaling pathways, could be monitored or even controlled simply by treating the live cells with chemical reagents and subsequently imaging.

We discussed this in page 10-11, line 171-184.

10) *Throughout the manuscript, italic "N" is used in GlcNAc, N-glycan etc. These should be non-italic.*

11) *In lines 85 & 86 "NAc-glucosamine" and "NAc-neuraminic acids" are not the official nomenclature for these monosaccharides.*

12) *Line 26, "a great deal of attention" should be re-phrased.*

13) *Line 88, "through an appropriate linker" should be re-phrased to a more specific sentence.*

14) *Line 105, "...the cyclic RGDyK ligand (...) could image all cells..." should be re-phrased to "...could be used to image all cells..."*

15) *Line 148, "biorthogonal" was probably autocorrected from "bioorthogonal"*

16) *Line 155, "to a greater extent" should be re-phrased and contain the fold increase of labeling.*

17) *Line 155, "by injection of injecting" word duplicated.*

18) *Line 193, "Scare bar" should be "Scale bar".*

19) *Line 241, "Hoechest" should be "Hoechst".*

We rephrased and corrected all typos, as kindly suggested.

Referee: 2

1) *I would like to see the expression levels of the cell surface proteins and compare them to the labeling efficiency. Although unexpectedly low labeling could be due to spatial arrangement, protein expression levels could be playing a role in labeling efficiency. One may consider that spatial arrangement may also have less effect if the labeling is performed at room temperature instead of 4°C, i.e., the cell-labeling profiles may be temperature dependent.*

We responded to the same comment by reviewer 1-6) above. Concerning on the temperature effects on spatial arrangements of two receptors, we have throughout optimized the conditions for pretargeting and

on-cell click reaction (*Adv. Sci.*, **2017**, 1700147); when the click reaction was performed at room temperature, the pretargeted RDG rather internalized and the efficiency of on-surface click reaction significantly reduced. We commented this in page 7, line 105-107.

2) On the other hand, a combined use of cyclic RGDyK peptides and N-glycan ligands resulted in specific cell labeling...Gal- and GlcNAc-terminated glycans (2c,d) on BxPC3 cells (Fig. 3c),...

- 2e should be added as well.

3) ... $\alpha(2,6)$ Sia- and $\alpha(2,3)$ Sia-terminated glycans (2a,b) on normal TIG3 cells (Fig. 3f).

- 2d,e should be added as well.

We corrected the description.

4) Is spatial arrangement the reason that we don't see binding of 2e to A549 or is it protein levels? Should additional linker lengths be explored? Why does 2c appear to bind to PC3?

As pointed out by this reviewer, based on the Table 1, A549 express the SP-D, but the mannose-terminated glycan **2e** could not image the cells. Alternatively, the galactose-terminated glycan **2c** slightly stained the PC3 cells regardless of the linker length of RGD derivatives **1**, while no representative mannose-binding lectins are expressed on the PC3 cell surface. As this reviewer notes, this could be rationalized by the spatial arrangement and distance between the integrin and lectins on cell surface. In addition, the cell surface binding of glycans, especially when the interaction is very weak, is not only caused by the glycan/lectin interaction, but also by the hydrophobic or H-bonding interactions with various biomolecules on cell surface. The multiple interactions by the glycan on the cell surface complicate the analysis and cannot simply rationalized by the lectins expressed on the cell surface. We cannot simply rely on the glycan/lectin correlation data in Table 1, but this Table can be considered as one of the directions for analysis. We discussed this in page 11, line 185-194.

5) TIG3 shows strong staining with 2a,b but only has Vimentin. BxPC3 also expresses Vimentin, but is not stained with 2a,b. Vimentin is a mesenchymal marker and its expression level is dependent on the epithelial or mesenchymal characteristic of the cell. One may deliberately induce epithelial-to-mesenchymal transition, verify upregulated Vimentin, and test this pre-targeting technique again.

Thank you for the stimulating comments and constructive information. To the best of our knowledge, in most cases, Vimentin is reported to bind to the non-reducing end galactose or galactosamine (as shown in Table 1). While the possibility of binding to N-sialoglycans, **2a,b**, cannot be ruled out for special occasion, we expect that the binding of the N-sialoglycans with TIG3 could be due to the unknown lectins or other interactions on cell surface.

We hope that these revisions have dealt with in satisfactory manner. As usual we are most grateful for the referee's constructive comments as well as your services.

Reviewers' comments:

Reviewer #1 (Remarks to the Author):

The issues raised previously have been resolved. This is an interesting piece of work that should be the basis for further extension, especially mechanistically (why is expression of cellular "receptors" not consistent with labelling?). But this is out of the scope of this paper.

The manuscript should be ready for publication as soon as, in the interest of reproducibility, the authors state which mycoplasma test they are using for screening.

Reviewer #2 (Remarks to the Author):

Thank you for addressing our concerns. I would like to add new comments.

1) The new sentence "When the click reaction was performed at room temperature, the pretargeted RDG rather internalized and the efficiency of on-surface click reaction significantly reduced" has some problems. Please keep nomenclature consistent within the manuscript for clarity. I assume "pretargeted RDG" refers to the moieties mentioned in line 101 "pre-targeted cyclic RGDyk." I think the authors are trying to say, "At room temperature, the higher rate of integrin-mediated internalization of the pre-targeted cyclic RGDyk reduced the efficiency of the on-surface click reaction when compared to the 4°C reaction conditions." Also, I feel "pre-targeted" describes the following noun, in this case, the peptide. For example, a pre-targeted approach or pre-targeting approach would refer to a process that contains a pre-targeting step. Although I do see the term "targeted peptide" in the literature, "pre-targeted cyclic RGDyk" means the peptide, not the integrin or cell, is the molecule being (pre-)targeted. "Integrin-targeted cyclic RGDyk" would be more correct, I think. However, "target (cyclic RGDyk) peptide" or "targeting cyclic RGDyk" means the peptide is directing the integrin or click reaction.

4) If non-amenable spatial arrangement and distance are the explanation for the lower than expected staining, increasing the temperature of the click reaction conditions may increase the cell membrane fluidity to increase numbers or ranges of spatial arrangements and distances that could be sampled by the protein-ligands complexes during the same duration of time. Of course, the increased internalization rate would have to be considered again.

5) Could you use 2a,b to pull down and identify the interacting molecule?

For the new sentences:

Line 63: "As we have investigated in the previous research,⁸ the clicked products efficiently internalized with the same efficiency as RGD peptide, while glycans hardly internalized."

Proposed revision: As we have investigated in the previous research,⁸ the clicked products are efficiently internalized with the same efficiency as the RGD peptide, while the glycans are less efficiently internalized.

Line 65: Because the click reaction should occur on the cell surface, click ligation and subsequent internalization could be the key of the method to image the target cells.

I don't understand this sentence.

Line 163: replace "big" with "large"

Line 193: "The multiple interactions by the glycan on the cell surface complicate the analysis and cannot simply be rationalized by the lectins expressed on the cell surface."

"The multiple interactions by the glycan on the cell surface complicate the analysis and the results cannot simply be rationalized by the lectins expressed on the cell surface."

Referee: 1

1) *The manuscript should be ready for publication as soon as, in the interest of reproducibility, the authors state which mycoplasma test they are using for screening.*

Thank you for the comment. We mentioned the source of the kit in page 21, line 278-279.

Referee: 2

1) *The new sentence “When the click reaction was performed at room temperature, the pretargeted RDG rather internalized and the efficiency of on-surface click reaction significantly reduced” has some problems. Please keep nomenclature consistent within the manuscript for clarity. I assume “pretargeted RDG” refers to the moieties mentioned in line 101 “pre-targeted cyclic RGDyK.” I think the authors are trying to say, “At room temperature, the higher rate of integrin-mediated internalization of the pre-targeted cyclic RGDyK reduced the efficiency of the on-surface click reaction when compared to the 4°C reaction conditions.” Also, I feel “pre-targeted” describes the following noun, in this case, the peptide. For example, a pre-targeted approach or pre-targeting approach would refer to a process that contains a pre-targeting step. Although I do see the term “targeted peptide” in the literature, “pre-targeted cyclic RGDyK” means the peptide, not the integrin or cell, is the molecule being (pre-)targeted. “Integrin-targeted cyclic RGDyK” would be more correct, I think. However, “target (cyclic RGDyK) peptide” or “targeting cyclic RGDyK” means the peptide is directing the integrin or click reaction.*

Yes, the reviewer is absolutely right. Thanks to the comments, we put the appropriate sentences as kindly suggested. In addition, “pre-targeted cyclic RGDyK” was replaced by “Integrin-targeted cyclic RGDyK” or “on-cell cyclic RGDyK”.

4) *If non-amenable spatial arrangement and distance are the explanation for the lower than expected staining, increasing the temperature of the click reaction conditions may increase the cell membrane fluidity to increase numbers or ranges of spatial arrangements and distances that could be sampled by the protein-ligands complexes during the same duration of time. Of course, the increased internalization rate would have to be considered again.*

As kindly advised, we performed the imaging of A549 cells by treating with **2b** or **2e** at the elevated temperature, instead of performing the click reaction at 4°C. We still cannot see any fluorescence signal by the treatment with **2e**, but in addition, the fluorescence intensity was significantly reduced even for the case of **2b** (see new Fig. S16). As analyzed by confocal microscopy, the treatment of A549 cells with TAMRA-labeled cyclic RGDyK at room temperature notably accelerated the internalization and accumulated in the nuclei (see new Fig. S17). These additional experiments clearly showed that the higher rate of integrin-mediated internalization of the on-cell cyclic RGDyK reduced the efficiency of “on-cell” click reaction, as discussed previously. We totally agree with this reviewer’s point that “increasing the temperature of the click reaction conditions may increase the cell membrane fluidity”, so that reactivity and fluorescence intensity should increase. But unfortunately, we could not see the effects due to the internalization property of Integrin/cRGDyK complex. We commented this in SI.

5) *Could you use 2a,b to pull down and identify the interacting molecule?*

As discussed previously, the “weak” glycan interacting is not necessarily caused by the surface lectins, but also with membrane proteins, glycans, lipids, or even by the hydrophobic or H-bonding interactions

on cell surface. That is quite complex. While we admit that identifying and analyzing the glycan-interacting molecules are important, that is not the main point of this study. By combining with the “strong” integrin/RGD cell surface interaction, we made the cell-identifying fingerprints profile. These fingerprints could be used to selectively image the targeted cancers even in xenografted mice, which may provide quite promising directions for future cell targeting or imaging by using the “weak” glycan interaction.

For the new sentences:

Line 63: “As we have investigated in the previous research,⁸ the clicked products efficiently internalized with the same efficiency as RGD peptide, while glycans hardly internalized.” Proposed revision: As we have investigated in the previous research,⁸ the clicked products are efficiently internalized with the same efficiency as the RGD peptide, while the glycans are less efficiently internalized.

Line 65: Because the click reaction should occur on the cell surface, click ligation and subsequent internalization could be the key of the method to image the target cells. I don’t understand this sentence.

Line 163: replace “big” with “large”

Line 193: “The multiple interactions by the glycan on the cell surface complicate the analysis and cannot simply rationalized by the lectins expressed on the cell surface.” “The multiple interactions by the glycan on the cell surface

We rephrased and corrected all typos, as kindly suggested.

We hope that these revisions have dealt with in satisfactory manner. As usual we are most grateful for the referee’s constructive comments as well as your services.

Yours sincerely,

Dr Katsunori Tanaka
Chief Scientist
Biofunctional Synthetic Chemistry Laboratory
RIKEN Cluster for Pioneering Research
2-1 Hirosawa, Wako-shi, Saitama 351-0198, Japan
E-mail: kotzenori@riken.jp
Tel: +81-48-467-9405, Fax: +81-48-467-9379

REVIEWERS' COMMENTS:

Reviewer #2 (Remarks to the Author):

Dear Editor,

The authors have addressed my comments well with their responses and additional data.

I have very minor corrections to suggest:

line 51:

"dibenzocyclooctyne (DBCO), which is bioorthogonal to azide" to "dibenzocyclooctyne (DBCO), which reacts bioorthogonally with azide"

line 165:

"largely" to "primarily". (my mistake, as I suggested my default alternative adjective without considering the resulting sentence)

line 175:

change i.e, to i.e.,

Supplementary:

Page 8, paragraph 2, line 3:

change "rified" typo to "purified"

Section 3.1 and 3.2

I assume eight-ten weeks old mice means eight 10-week-old mice

Referee: 2

line 51:

"dibenzocyclooctyne (DBCO), which is bioorthogonal to azide" to "dibenzocyclooctyne (DBCO), which reacts bioorthogonally with azide"

line 165:

"largely" to "primarily". (my mistake, as I suggested my default alternative adjective without considering the resulting sentence)

line 175:

change i.e, to i.e.,

Supplementary:

Page 8, paragraph 2, line 3:

change "rifed" typo to "purified"

Supplementary:

Section 3.1 and 3.2

I assume eight-ten weeks old mice means eight 10-week-old mice

We rephrased and corrected all typos, as kindly suggested.

We corrected the phrase "eight -ten weeks old mice" to "Mice (from 8 to 10 weeks old)".